# Analysis of the Development Status and Prospect of China’s Agricultural Sensor Market under Smart Agriculture

**DOI:** 10.3390/s23063307

**Published:** 2023-03-21

**Authors:** Ning Sun, Beibei Fan, Yahui Ding, Yuanjinsheng Liu, Yuyun Bi, Patience Afi Seglah, Chunyu Gao

**Affiliations:** 1Institute of Agricultural Resources and Regional Planning, Chinese Academy of Agricultural Sciences, Beijing 100081, China; 2National Engineering Research Center for Information Technology Research in Agricultural, Beijing 100097, China; 3School of Science, China University of Geosciences (Beijing), Beijing 100083, China

**Keywords:** smart agriculture, agricultural sensors, development prospects, existing problems

## Abstract

Agricultural sensors are essential technologies for smart agriculture, which can transform non-electrical physical quantities such as environmental factors. The ecological elements inside and outside of plants and animals are converted into electrical signals for control system recognition, providing a basis for decision-making in smart agriculture. With the rapid development of smart agriculture in China, agricultural sensors have ushered in opportunities and challenges. Based on a literature review and data statistics, this paper analyzes the market prospects and market scale of agricultural sensors in China from four perspectives: field farming, facility farming, livestock and poultry farming and aquaculture. The study further predicts the demand for agricultural sensors in 2025 and 2035. The results reveal that China’s sensor market has a good development prospect. However, the paper garnered the key challenges of China’s agricultural sensor industry, including a weak technical foundation, poor enterprise research capacity, high importation of sensors and a lack of financial support. Given this, the agricultural sensor market should be comprehensively distributed in terms of policy, funding, expertise and innovative technology. In addition, this paper highlighted integrating the future development direction of China’s agricultural sensor technology with new technologies and China’s agricultural development needs.

## 1. Introduction

The emergence of smart agriculture (SA) correlates with the rapid development of information technology in the 21st century. SA is a new mode of agricultural production that emphasizes information and knowledge as its fundamental components. Incorporating technologies such as the Internet, the Internet of Things (IoT), big data, cloud computing and artificial intelligence into agriculture allows for the collection and analysis of agricultural information and they aid in decision-making based on quantitative data. Given this, SA has major significance for developing the agricultural sector [1].

As an important front component of SA, agricultural sensors convert non-electrical physical quantities (internal and external factors) related to animals and plants into electrical signals recognized by control systems. Sensors are used to collect data regarding physical and environmental attributes, whereas actuators are used to respond to feedback and gain control over conditions. The sensors’ accumulated data about the object or environment can be used to identify people, locations, objects and their states, which is a guiding principle for scientific research [2]. Agricultural sensors are the foundation of agricultural IoT technology and SA [3]. They are also the basis for the digitalization of the agricultural industry [4]. In this regard, developing and utilizing agricultural sensors is vital for developing SA and establishing agricultural and rural big data systems [5]. The sensors can effectively reduce labor costs and increase the survivability and productivity of plants and animals. Similarly, they enhance labor productivity, improve resource utilization and land productivity and overcome environmental and resource constraints on agricultural development [6].

According to the detection objects, agricultural sensors can be classified into environmental, animal, plant, agricultural product and agricultural machinery sensors. By employing the IoT and big data, agricultural sensors can lessen the ecological difficulties associated with traditional agriculture in open-field crop cultivation, enclosed crop cultivation, animal husbandry and aquaculture production.

According to publicly available data, the global market for agricultural sensors was valued at USD 1.23 billion in 2018 and is expected to reach USD 2.56 billion in 2026, growing at a compound annual growth rate (CAGR) of 11.04%. In 2025, the adoption rates of digital technology in China’s open-field crop cultivation, enclosed crop cultivation, animal husbandry and aquaculture industries will reach 25%, 45%, 50% and 30% [4]. Furthermore, in 2035, the adoption rates of digital technology in China’s open-field crop cultivation, enclosed crop cultivation, animal husbandry and aquaculture industries will reach 50%, 70%, 75% and 75% [6]. As a traditional agricultural country, China is transitioning to smart agriculture. There is a considerable demand for agricultural sensors, which are essential for SA. The development of China’s agricultural sensor industry, on the other hand, is confronted with opportunities and challenges.

The United States, Japan and Germany hold the largest market share in the global sensor market distribution, whilst other countries, including China, lag behind with a significant gap in market share [7]. For instance, developed countries, including those in Europe, the United States and Japan, monopolize airborne-to-earth observation core sensor technology. In addition, countries such as the United States, Japan, the United Kingdom, France and Germany have made sensor technology one of their national priorities for developing critical technologies. This has contributed to the rapid development of sensor technology [8]. At present, China’s agricultural sensor field is facing many problems such as a lack of enterprise technology innovation capability and key technologies, international competitiveness and unsound industrial ecology [9]. Almost 100% of the medium- and high-quality sensing components and 90% of the sensor chips used in China are imported [10]. Due to a lack of high-quality optical components, high-end sensor chips and other principal technologies and processes, the development of airborne spectral imaging sensors is still in its infancy in China [11]. The research and development of multispectral camera sensors in China are only in the scientific research stage. In this regard, users of agricultural UAV sensors can only use expensive imported multispectral sensors. Similarly, wearable plant water sensors [12], electronic noses for monitoring volatile organic compounds released from plant leaves [13,14], miniature biosensors for detecting pH changes in plant foliar tissues [15], nanosensors for detecting crop stress signaling molecules [16] and other plant and animal life ontology information sensors have a long way to go before they can be widely adopted in China. Chinese agricultural sensors, particularly intelligent agricultural sensors, are heavily dependent on imports. Meanwhile, the production and application of self-produced agricultural sensor core components are insufficient to meet the rapid development of the agricultural IoT and SA [17]. This has become a bottleneck that restricts the growth of SA in China.

Through a comprehensive literature review and data statistics, this paper analyses the current situation and prospects of the development of agricultural sensors in China. Likewise, the present study points out the problems and future development directions of China’s agricultural sensor industry in an attempt to serve as a reference for developing agricultural sensors and smart agriculture in China.

## 2. State-of-the-Art and Related Work

According to the agricultural sensor market research report, the global agricultural sensor market will reach CNY 37.793 billion in 2021. The report predicts that by 2027 the global agricultural sensor market will reach CNY 111.453 billion at a compound annual growth rate of 19.75% during the forecast period. Agricultural sensors have expanded into a vast sector with a wide range of applications in response to the different characteristics of agricultural production. In general, agricultural sensors can be divided into two main categories depending on the object of detection—namely, environmental information sensors and life information sensors [18,19,20]. A more in-depth division of agricultural sensors includes agricultural–environmental information sensors, plant and animal life information sensors, agricultural product quality and safety information sensors, agricultural machinery working condition sensors and operation sensors. In SA production and applications, agricultural sensors can be subdivided into four categories: soil sensors [21], water sensors [22], meteorological sensors [23] and sensors integrated into livestock breeding environments [24]. 

The soil sensors detect temperature, humidity, pH values, chemical fertilizer concentrations and pesticide concentrations [25]. Soil sensors play a critical role in SA by monitoring real-time physical and chemical signals in the soil. They optimize crop growth circumstances, fight against biotic and abiotic stresses and enhance crop yields. At present, there are many advanced soil sensors in the Chinese market. In recent years, agricultural sensors developed based on microelectronic system (MEMS) technology have been popularized and applied in agricultural science and technology due to their economic, accurate and sensitive characteristics. Using advanced sensors and information technology has become a method to improve the productivity and quality of modern agriculture [26]. The soil sensor market is huge. Among it, the global market for soil moisture sensors was estimated at USD 147.5 million in 2020 and is projected to reach USD 360.9 million by 2027 [27].

Regarding water sensors, the parameters are dissolved oxygen, temperature, pH values, salinity, conductivity and ammonia concentrations. In the future, developing low-cost water sensors with minimum power consumption, strong practicability, comprehensive network coverage and adequate data visualization is the main direction for developing water sensors [28]. The use of meteorological sensors has been well-established. In addition to traditional parameters such as light intensity, temperature, humidity, carbon dioxide concentration and soil moisture content [29], parameters such as aerosol accumulation and carbon loss have been detected by meteorological sensors [30]. With the application of wireless sensor networks [31], the intelligent calibration method [32] and multi-thread serial communication technology [33] on sensors, agrometeorological sensors are now more accurate and cheaper. At last, temperature, humidity, ammonia concentrations, hydrogen sulfide concentrations, carbon dioxide concentrations, sulfur dioxide concentrations and light intensity are the main parameters detected by environmental sensors integrated into breeding environments for poultry and livestock [34]. Environmental sensors should ideally be affordable, lightweight, modular and portable [35]. Currently, more wearable sensor technologies such as motion sensors and vital signs sensors are used within the breeding environment to collect, transmit and store information [36].

In addition, more sensors are used to monitor crops and animals’ growth and life information in real time. Some of these sensors are used to realize the autonomous navigation of agricultural machinery, support scientific decision-making and help agricultural production. Smart optical sensors (automated and high-precision) for detecting the chlorophyll content of a leaf [37], the moisture content of a leaf [38] and water transport in plants [39] have been developed. Animal sensors mainly monitor animals’ growth and physiological parameters (body temperature, behavior, respiratory rates and blood pressure). Animal sensors based on wireless communication modules, infrared light and radio frequency identification have been adopted to identify sheep, dairy cows and sows’ physiological states (diseased, estrous, grazing and feeding) [40,41]. The laser sensor is a new sensor that can detect target shape characteristics without making contact, perform autonomous agricultural machinery avoidance activities, increase irrigation water use efficiency and increase land flatness [42]. The vision sensor simulates human vision to extract usable information from the acquired image, interpret and comprehend it and then utilize it to control agricultural machinery or equipment. Computer and pattern recognition technology is used in this process [8,43].

## 3. Materials and Methods

According to the classification method widely used in China, this study divided China’s agricultural sensor market into four categories: open-field farming sensors (environmental information sensing sensors and farm machinery sensors), facility farming sensors (environmental information sensing sensors and plant life information sensors), livestock and poultry farming sensors (pig life sensors only) and aquaculture sensors (freshwater farming). 

A literature search was carried out by searching the “China National Knowledge Infrastructure (CNKI)”, “Web of Science”, “Springer Link” and “Science Direct” with the keywords of “smart agriculture”, “agricultural sensors”, “agricultural sensor market”, “agricultural sensor industry”, “development prospects”, “existing problems” and “China”. The selection of articles was made based on the keywords mentioned above and included the following criteria: (1) articles related to agricultural sensors and sensor technology, (2) data and forecast on the application of agricultural sensors in China and (3) analysis of the prospect of China’s agricultural sensor market and existing problems. The search results were limited to publications in English and Chinese, and priority was given to articles published during 2015–2022. In addition, this paper also searched government documents and official websites about agricultural sensors in China and consulted documents related to SA and agricultural sensors (Table 1).

## 4. Results

This section analyzes agricultural sensors’ development status and prospects in China, from field planting to facility planting, livestock and poultry breeding and aquaculture.

### 4.1. The Demand from the Open-Field Crop Cultivation Industry for Agricultural Sensors

According to data from a survey conducted in 2020 involving 613 farmers in China, there is a strong demand for soil and agricultural machinery sensors for the operations of the open-field crop cultivation industry. Currently, the main types of sensors applied in open-field crop cultivation are environmental and agricultural machinery sensors. The environmental sensors include soil sensors, meteorological sensors and water sensors. On the other hand, agricultural machinery sensors include fuel consumption, speed, GPS, angle and attitude sensors. However, according to the 2020 National County Digital Agriculture and Rural Development Level Evaluation Report, the adoption rate of information technology in China’s open-field crop cultivation industry in 2019 was 17.4%, indicating that environmental and agricultural machinery sensors are not widely applied.

For environmental monitoring, a small soil moisture monitoring station integrated with an air temperature sensor, an air humidity sensor, a soil temperature sensor, a soil moisture sensor and a soil pH sensor can cover up to 10 mu of land. According to the “red line” policy, the total arable land in China should be no less than 1.8 billion mu (120 million hectares (ha)). Therefore, the potential demand from the open-field crop cultivation industry for environmental sensors is as high as 900 million units, corresponding to a market size of CNY 450 billion. It is estimated that the adoption rates of digital technology in China’s open-field crop cultivation industry will exceed 25% and 50% in 2025 and 2035. The demand from the open-field crop cultivation industry for environmental sensors is expected to reach 225 million units at the end of 2025 and 450 million units in 2035.

In China’s agricultural industry, tillage, seeding, pipe installation and harvesting machines have sensors to monitor their operation. The machines include precision rice seeders, no-tillage precision seeders, precision plant protection machines and precision corn harvesters [19,44]. According to the Ministry of Industry and Information Technology of the People’s Republic of China (PRC), from January 2020 to June 2020, the total automated agricultural machinery sales generated by major agricultural machinery manufacturers, including China Yituo Heavy Industry Co., Ltd., Revo Heavy Industry Co., Ltd. and Zoomlion Heavy Industry Co., Ltd., exceeded 11,700 units, corresponding to annual growth of 213%. In addition, the China Statistical Yearbook 2021 highlighted that, in 2020, the total number of large- and medium-sized tractors in China was 4.47 million. In 2018, the total market volume of large- and medium-sized tractors in China decreased significantly. Notwithstanding, the production and sales of Chinese tractor enterprises continued to decline due to numerous factors. These factors included large social ownership, the extension of the investment return period, reduced planting income, increased user demand power and other factors that restrained market demand. Due to the influence of policies (such as subsidies, emission limits and land consolidation) on the ownership of large- and medium-sized tractors in China, no mathematical equation can reliably predict the development accurately [45]. Therefore, a generalized linear model, an index model and a first-order autoregressive model were used to predict the number of large- and medium-sized tractors and the demand for agricultural machinery sensors in China in 2025 and 2035. The results are shown in Table 2, Table 3 and Table 4 and Figure 1. 

As shown in Table 3, all of the parameter estimates obtained by using the three regression models are significant (*p*-values < 0.001), but the coefficient of determination (R^2^ value) of the first-order autoregressive model is higher than the R^2^ values of the generalized linear model and index model.

As shown in Table 4, the average relative error of values predicted by the first-order autoregressive model is lower than the average relative error of values predicted by the generalized linear and index models. Therefore, the first-order autoregressive model predicts the number of large- and medium-sized tractors in China more accurately than the generalized linear and index models. The equation for the first-order autoregressive model is as follows: $$y_{t+1}=0.8948y_{t}+569139$$
where $y_{t+1}$ represents the number of large- and medium-sized tractors in China in the year (t + 1) and $y_{t}$ represents the number of large- and medium-sized tractors in China in year t. 

Based on the changes in the number of large- and medium-sized tractors in China from 2001 to 2020 (Table 2), it is expected that the number of large- and medium-sized tractors in China will reach 4.71 million and 4.98 million in 2025 and 2035. An automated wheat harvester requires a set of position, vision and measurement sensors. In total, at least 30 sensors are required in an automated wheat-harvesting process to configure the data collection process and synchronize the collected data. Accordingly, as shown in Table 5, the demand from the open-field crop cultivation industry for agricultural machinery sensors will exceed 35.23 million units and 74.70 million units in 2025 and 2035, respectively. The potential market size for agricultural machinery sensors will be CNY 66.4 billion. 

### 4.2. The Demand from the Enclosed Crop Cultivation Industry for Agricultural Sensors

Enclosed crop cultivation, also known as “facility agriculture”, refers to agricultural production that adopts enclosing facilities such as plastic greenhouses (made of glass or polycarbonate boards), solar greenhouses and awnings. It is commonly used to cultivate vegetables, fruits and flowers. Currently, the primary sensors applied in enclosed crop cultivation include environmental and plant sensors. The environmental sensors comprise temperature, humidity, soil moisture and carbon dioxide sensors. On the other hand, plant sensors constitute sensors for monitoring plant growth, nutrients, hormones, diseases and pests [46]. These seven integrated sensors are air temperature sensors, humidity sensors, light intensity sensors, soil moisture sensors, soil pH sensors, soil nutrient sensors and carbon dioxide sensors.

In 2018, the Ministry of Agriculture and Rural Affairs (MARA) and the Ministry of Natural Resources of PRC launched a nationwide campaign to clean up illegal “greenhouses” and facilities that occupy farmland to accelerate the development of enclosed crop cultivation. In 2020, the General Office of the State Council of PRC issued a notice on ”Resolutely Preventing the Conversion of Cultivated Land into Agricultural Land” and ”Notice and Opinions on Preventing the Conversion of Cultivated Land into Grain” to consolidate the results of the rectification of the illegal “greenhouses” and facilities. The notice also strengthened the supervision of lands used for agricultural facilities. In this regard, since 2019, the total area occupied by enclosed crop cultivation facilities has decreased. According to the China Statistical Yearbook 2020, at the end of 2019, the total area occupied by enclosed crop cultivation facilities was 3.04 million hectares (Figure 2). 

In comparison, the total area occupied by the facilities at the end of 2018 was 3.47 million hectares. According to the opinions of MARA on accelerating the development of facility planting mechanization, released in 2020, in China, the total area occupied by enclosed crop cultivation facilities (plastic greenhouses, solar greenhouses and connected greenhouses) will exceed 2 million hectares in 2025. Although the total area occupied by enclosed crop cultivation facilities has decreased in China since 2018, the adoption rate of digital technology in the country’s enclosed crop cultivation industry has increased. The adoption of digital technology in China’s enclosed crop cultivation industry is expected to increase from 45% in 2025 to 70% by 2035 [6]. Based on the requirement of a smart greenhouse occupying one mu of land for eight environmental sensors, the demand from the enclosed crop cultivation industry for environmental sensors will reach 108 million units in 2025 and increase to 168 million units in 2035 (Table 6). The eight environmental sensors mentioned above are air temperature sensors, humidity sensors, light intensity sensors, soil moisture sensors, soil pH sensors, soil nutrient sensors and carbon dioxide sensors. About 54 million and 84 million plant sensors will be needed to meet industry demand by 2025 and 2035, respectively. For context, the environmental sensors market is estimated to be worth CNY 54 billion, while the plant sensors industry is estimated to be worth CNY 27 billion. 

### 4.3. The Demand from the Animal Husbandry Industry for Agricultural Sensors

The animal husbandry industry in China is developing from a traditional industry into an intelligent, digitalized and centralized industry. Environmental and animal sensors are the two most prevalent sensors utilized in the animal husbandry sector. Installing agricultural sensors in livestock sheds, poultry houses and animal bodies is essential to the intelligent animal husbandry industry. They allow for the automated monitoring and control of farming environments and the collecting, storing and analyzing of data related to farming processes. The 526 animal husbandry enterprises surveyed in 2020 revealed that 54.55% had automated monitoring of breeding environments. The monitored parameters were traditional environmental parameters such as temperature, humidity, light intensity and ventilation. Considering the 2020 National County Digital Agriculture and Rural Development Level Evaluation Report, in 2019, the adoption rate of digital technology in livestock and poultry breeding in China was 41%. In this regard, the adoption rates of digital technology in China’s animal husbandry industry are anticipated to reach 50% in 2025 and 75% in 2035 [6]. This signifies that China’s animal husbandry industry has fast forwarded to informatization and digitalization, increasing the demand for sensors used to monitor breeding environments for poultry and livestock.

In the swine-breeding industry, a hog house is usually equipped with at least five environmental monitoring sensors, including an air temperature sensor, an air humidity sensor, a carbon dioxide sensor, an ammonia sensor and a light intensity sensor. The China Agricultural Outlook (2020–2029), released by MARA, alludes that in the next 10 years China’s pork output will grow at an average annual rate of 1.9%. Moreover, the supply and demand for live hogs will approach equilibrium in 2035. The number of slaughter hogs is expected to reach 347 million in 2025 and will stabilize at 374 million in 2035. Based on the assumption that 20 hogs occupy one hog house, the potential size of the market for environmental sensors used in the swine-breeding industry is CNY 41.2 billion. The demand from the industry for environmental sensors will reach 43.37 million units in 2025 and 70.12 million units in 2035. In addition to environmental sensors, animal sensors are crucial in monitoring breeding sows. Therefore, breeding sows are usually equipped with wearable vital-sign monitoring equipment consisting of four sensors: a respiratory rate sensor, a body temperature sensor, a pulse rate sensor and a blood pressure sensor [47]. China had 41.61 million breeding sows as of the end of December 2020. Projections from the Opinions on Promoting the Sustainable and Healthy Development of the Pig Industry, which were issued by MARA, the National Development and Reform Commission (NDRC) of the PRC and six other departments, state that the number of breeding sows in China will stabilize at 43 million by the end of 2025. The demand from the swine-breeding industry for animal sensors (including individual identification sensors, animal behavior sensors, animal physiology sensors and animal disease sensors) is predicted to reach 86 million units in 2025 and 129 million units in 2035 (Table 7). Similarly, the potential market size for the sensors is CNY 86 billion (Table 7).

### 4.4. The Demand from the Aquaculture Industry for Agricultural Sensors

China’s aquaculture industry is one of the fastest developing in the country’s agricultural sector. In China, aquatic products come mainly from the country’s aquaculture industry. According to the China Fishery Statistical Yearbook 2020, China’s national aquaculture output in 2019 was 51 million tons, corresponding to year-on-year growth of 1.76%. On the other hand, the total area designated for aquaculture production was 7 million ha, corresponding to a yearly decrease of 1.13%. In 2016, MARA issued working specifications for the preparation of an outline of the flat tidal planning in aquaculture waters. The documents put forward zoning and classification guidelines and supporting policies related to the distribution of aquaculture production. In addition, in 2018, it was proposed that adjustments to China’s aquaculture industry should focus on “improving quality and reducing quantity” and “improving the ecological aspects of aquaculture” [48]. China’s aquaculture industry needs industrial restructuring and intelligent transformation due to the slow growth of output of aquatic products and the area designated for aquaculture production. At present, the adoption rate of digital technology in China’s aquaculture industry is low because most enterprises and farmers still adopt traditional aquaculture production. The 2020 National County Digital Agriculture and Rural Development Level stipulates that, in 2019, the adoption rate of digital technology in China’s aquaculture industry was 16.4%. However, this value was lower than the adoption rates of technology in the country’s animal husbandry, open-field crop cultivation and enclosed crop cultivation industries. Only a small fraction (>1%) of China’s counties (cities and districts) adopted digital technology in their aquaculture farms. Thus, China’s agriculture sector is still in the early stages of its intelligent transformation. There is growing consensus that intelligent transformation is key to developing the modern aquaculture industry. Sensing technology and the IoT enable innovative transformation through comprehensively monitoring and controlling water quality and the growth of cultivated species in aquaculture production environments. The technologies will continuously improve the productivity and profitability of the aquaculture industry [49]. Therefore, there is a great demand from the aquaculture industry for agricultural sensors.

Currently, in the aquaculture industry, sensors are mainly used to monitor the pH and temperature of the water and the dissolved oxygen and ammonia concentrations in water. Based on the standard coverage of the sensors, a pH sensor, a dissolved oxygen sensor and a temperature sensor are required to cover an area of 10 mu. At present, galvanic dissolved oxygen sensors are most commonly (50%) used in the aquaculture industry, followed by polarographic (40%) and optical (10%) dissolved oxygen sensors [50]. The average maintenance periods of the galvanic, polarographic and optical dissolved oxygen sensors are three months, six months and twelve months, respectively. In contrast, composite glass electrode pH sensors are the pH sensors most commonly (70%) used in the aquaculture industry, followed by differential (28%) and optical (2%) pH sensors. The average maintenance periods of the composite glass electrode, differential and optical pH sensors are one month, three months and six months, respectively [50]. The aquaculture industry generally has a high demand for the three types of dissolved oxygen and pH sensors. The sensors have average maintenance intervals of less than one year. In terms of the area designated for aquaculture production, due to the continuous promotion of the supply-side reform of national aquaculture and the increasing awareness of environmental protection, the aquaculture production area in China will remain stable. China’s total area used for freshwater aquaculture production in 2019 was 5.12 million hectares. Considering the assumption that the coverage of an area of 10 mu requires the installation of a pH sensor, a dissolved oxygen sensor and a temperature sensor, the demand from China’s freshwater aquaculture industry for agricultural sensors will reach 9.2 million units (in 2025) and 23 million units in 2035 (Table 8). Additionally, the potential market size for the sensors is CNY 15.2 billion (Table 8). The adoption rates of digital technology in China’s aquaculture industry are expected to reach 30% in 2025 and 75% in 2035. 

## 5. Discussions

China’s agricultural sensor industry was established in the 1960s [51]. Supported by the rapid development of remote sensing technology, the industry gradually developed. At the end of the 1980s, remote sensing data acquisition using multiband sensors on satellites was established in China to monitor and evaluate agricultural production. At the same time, the connected greenhouse structure was introduced, and the scale of enclosed crop cultivation began to expand. Agricultural researchers and practitioners of enclosed crop cultivation began to use sensors in greenhouses and explore the applications of sensors in the agricultural industry. Over the past 40 years, China’s agricultural sensor industry has transformed into a developed industrial system [51]. Environmental sensors such as air temperature, air humidity and soil moisture sensors have been commonly used in agricultural production. Advanced technologies in environmental sensors focus on temperature, light, soil moisture, pH and other indicator measurements. However, there is a lack of sensors for the dynamic monitoring of the ecologically integrated environment of farmland and plant growth information. Research on dynamic sensing and monitoring technologies for harmful pollutants such as heavy metals and pesticide residues in soil, essential environmental factors and kinetic models for plant–soil–environment interactions is still insufficient. There is a lack of methods for highly sensitive, selective, multi-point simultaneous or multi-component high-throughput detection of the above single-component detection objects [18,52]. On the other hand, temperature sensors, ammonia sensors, dissolved oxygen sensors and pH sensors have been used in aquaculture production. Moreover, multifunctional sensors integrating different types of sensors have been developed. Entering China’s “14th Five-Year Plan” period, the agricultural sensor industry is provided with a significant opportunity for development due to the great demand for agricultural sensors. Despite the vital opportunity, China lags behind technologically advanced countries in research on agricultural sensors and market share in the global market for agricultural sensors. Therefore, comprehensive improvements are required.

In China, research on agricultural sensors is still in its early stage. China produces less than 10% of the world’s agricultural sensors due to a dearth of investment and qualified research and development personnel [53]. Most of the agricultural sensors produced in China are modified industrial sensors. Therefore, agricultural sensors lack accuracy, stability and stress resistance. Moreover, neither animal nor plant sensors are produced in China, meaning core algorithms for big data mining process data collected by high-tech agricultural sensors are lacking [54]. Autonomous, controllable, intelligent, high-tech, agricultural sensors have not been produced in China. Most high-tech agricultural sensors developed in China are still in the laboratory testing stage. Significant differences exist in the stability, reliability and power consumption between domestically produced and imported agricultural sensors [55]. Additionally, the industrial development mechanism of China’s agricultural sensor technology needs to be improved. Problems with the industrial structure of agricultural sensors include fragmentation of enterprises, weakness, low technology level, the existence of a large number of similar products and a lack of innovation. Few companies in China can independently develop and improve sensors, while the majority of sensor companies duplicate or assemble similar products from overseas. As a result, the core technology accumulation of these enterprises is insufficient and far behind the developed countries. For example, the cost of developing mainstream sensor chips based on MEMS technology is high. Even a large-scale professional, scientific research team needs at least ten years of technical accumulation to succeed in research. Small sensor enterprises find it challenging to devote time and financial resources to developing new technology.

However, the United States, Germany, Japan and other countries are leading in agricultural sensors. These countries have a monopoly on sensing components, high-end agricultural environmental sensors, plant and animal life information sensors and online inspection equipment for the quality of agricultural products and other related technology products. The United States is one of the pioneers in using agricultural sensors in the agriculture sector. Not only does agricultural sensor technology in the United States feature a sophisticated measurement system, but the obtained data are also diverse and accurate. Different types of agricultural sensors have the characteristics of high integration. Therefore, the stability of information transmission between agricultural sensors is relatively strong. In addition, the United States invests more money in researching new technologies and new products of agricultural sensors every year. For example, the United States is developing laser-induced spectroscopy technology for measuring soil nutrients and heavy metal content and is using micro- and nanotechnology to develop sensors that can enter the metabolic cycle system of plant and animal life forms [54]. China’s proportion of the global market for agricultural sensors is considerably less than that of the United States, Japan and Germany. Agricultural sensors produced in China are manufactured on a small scale with low-level technology. Therefore, the profitability of the sensors is low because of the lack of economies of scale. As a result, approximately 90% of imported high-tech agricultural sensors dominate the Chinese market for agricultural sensors [10]. 

To summarize, smart agriculture has become an essential way for developed countries to increase the export value of agricultural products. As the technical support for smart agriculture is missing, the industrialization level of agricultural sensors in China lags far behind that of developed countries, mainly including the following aspects. (a) Large differences in sensor accuracy, low-cost performance, high product failure rate and low informatization level. (b) A lack of sensor communication technology, such as wireless sensor network communication, heavily affects accuracy, reliability and coverage. (c) The measurement type is singular, and the multi-parameter and multi-field integrated sensor needs to be developed urgently. (d) A lack of sensor intelligent calibration technology and application in error analysis, modeling and processing. (e) A lack of comprehensive and reliable monitoring technology, including electromagnetic compatibility technology and comprehensive monitoring of environmental reliability and safety regulations. (f) Diversity is absent in the types of agricultural planting and breeding sensors. Additionally, the application of biological agriculture planting and breeding sensors based on nano- and microsystem technology must be urgently developed. (g) Sensor materials and processes, such as nanoscale materials and MEMS sensor technologies, lack technological accumulation. China will need to make significant advancements in agricultural sensor technology. First, future optical, electrochemical, electromagnetic, ultrasonic, image and other methods based on new mechanisms of agricultural sensing, as well as sensitive devices, photoelectric conversion, weak signal processing and other core components, will become the critical research directions of China’s agricultural sensors. Then, new agricultural sensors with micro and small sizes, high reliability, low power consumption, low cost and long lifetime under complex agricultural environments will also be the cardinal directions of sensor technology development in China. According to the analysis of the products exhibited in the International Industrial Exhibition and the technological development of the internationally renowned manufacturers, the development trends of sensor technology are diverse. These encompass digital compensation, networking, intelligent and multi-function composite technologies. China should adapt its policies, finances, human resources and other resources and technology to the agricultural sensor market to accomplish its goals. By doing so, China can hope to achieve its primary strategy of breaking other countries’ monopoly on technical products by developing a batch of agricultural sensors with high precision, low cost and increased stability.

## 6. Conclusions

Since agriculture is the backbone of any country, ensuring its sustainable growth over the years is necessary. The development of SA and sensor technology provides opportunities for the sustainable development of agriculture. Based on our literature review, data statistics and the development trend of China’s agriculture, this work predicts the market potential of China’s agricultural sensors, highlighting that China’s sensor technology and enterprise R&D strength cannot meet the needs of China’s sensor industry. The sensors produced in China have the limitations of single measurement type, poor accuracy, high failure rate and a lack of sensor communication technology support. Similarly, sensor enterprises lack technology accumulation in materials and technology and cannot break through technical barriers. 

The current study posits solutions for addressing the abovementioned problems. First, China’s government must prioritize agricultural sensors as a principal technology for rural revitalization and development and design top-notch industrial systems. Second, enterprises must establish a sensor industry chain conducive to technology accumulation, research and innovation. Third, regarding technological breakthroughs, future optical, electrochemical, electromagnetic, ultrasonic, image and other methods based on new mechanisms of agricultural sensing, MEMS sensors, digital compensation, networking, intelligent and multi-function composite technologies, as well as sensitive devices, photoelectric conversion, weak signal processing and other core components, will become the critical research directions of China’s agricultural sensors. In addition, this paper also proposes that agricultural sensors will gradually develop towards low cost, high stability and high intelligence outcomes, as well as portability and ease of operation. Future technical improvements are expected to address the limitations of agricultural sensors.

## Figures and Tables

**Figure 1 sensors-23-03307-f001:**
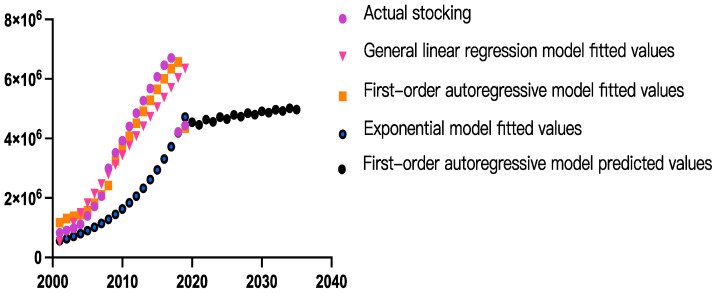
Fitting value and predictive value of ownership of large- and medium-sized tractors in China.

**Figure 2 sensors-23-03307-f002:**
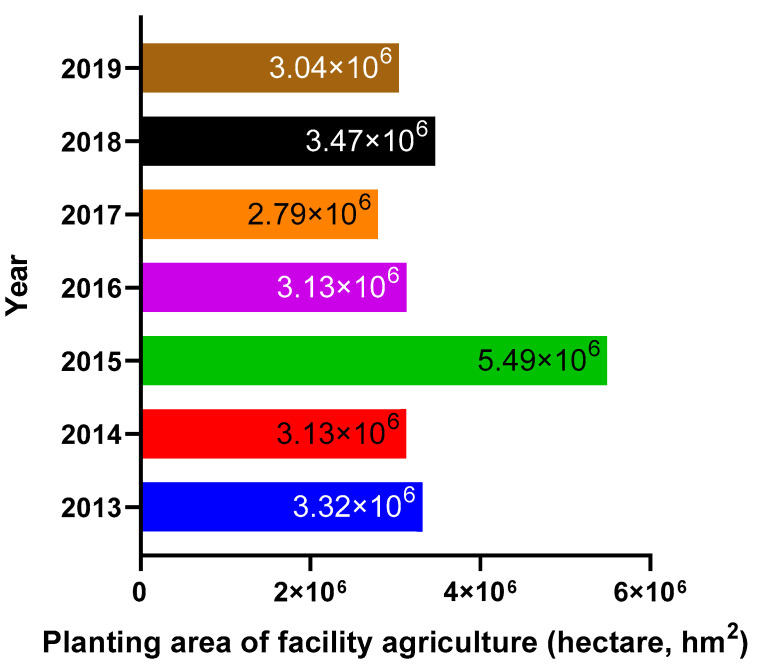
Changes in planting areas of facility agriculture in China from 2013 to 2019. Data source: China County Statistical Yearbook, National Bureau of Statistics.

**Table 1 sensors-23-03307-t001:** Government documents consulted.

Subject	Chinese National Government Entity	Documents Consulted
Smart Agriculture	Informatization Development Bureau of office of the Central Cyberspace Affairs Commission, Department of Market and Information Technology, Ministry of Agriculture and Rural Affairs of the People’s Republic of China, Chinese Bureau of Statistics	“No.1 Central Document” released by China in 2021, China Digital Rural Development Report (2020), statistical yearbook (2020)
Smart Agriculture	National Engineering Research Center for Information Technology Research in Agriculture	Research on Smart Agriculture Development Strategy for 2035
Agricultural Sensors	Animal Husbandry and Veterinary Bureau, Development Planning Division, Ministry of Agriculture and Rural Affairs of the People’s Republic of China	Summary of the Response Proposal No. 02409 (No. 195 for Agricultural and Water Conservancy) of the Fifth Session of the 13th National Committee of the Chinese People’s Political Consultative Conference (CPPCC), Response of the Ministry of Agriculture on Feasibility Study Report of Five Construction Projects including Precision Agriculture Technology Integrated Scientific Research Base (Fishery)

**Table 2 sensors-23-03307-t002:** Ownership of large- and medium-sized tractors in China from 2001 to 2020.

Year	Ownership of Large- and Medium-Sized Tractors	Year	Ownership of Large- and Medium-Sized Tractors
2001	829,900	2011	4,406,471
2002	911,670	2012	4,852,400
2003	980,560	2013	5,270,200
2004	1,118,636	2014	5,679,500
2005	1,395,981	2015	6,072,900
2006	1,718,247	2016	6,453546
2007	2,062,731	2017	6,700,800
2008	2,995,214	2018	4,219,893
2009	3,515,757	2019	4,438,619
2010	3,921,723	2020	4,472,700

Data source: China Rural Statistical Yearbook.

**Table 3 sensors-23-03307-t003:** Test results of three different models.

Model	Estimate Values	Standard Error	t Values	*p* Values	$R^{2}$ R Squares
General linear regression model	323,600.9	37,632.79	8.60	<0.001	0.8131
Index model	237.9 *	25.07	9.49	<0.001	0.8412
AR(1) autoregressive model	0.8948	0.1170	7.65	<0.001	0.8864

* The estimate here is the parameter estimate obtained by logarithmic transformation.

**Table 4 sensors-23-03307-t004:** Prediction errors of three different models.

Year	Actual Values *	General Linear Regression Model		Index Model		AR(1) Autoregressive Model	
		Predictive Values	Relative Error (%)	Predictive Values	Relative Error (%)	Predictive Values	Relative Error (%)
2016	6,453,546	5,379,616	16.64	3,311,698	48.68	6,003,170	6.69
2017	6,700,800	5,703,217	14.88	3,726,371	44.38	6,343,772	5.32
2018	4,219,893	6,026,818	42.81	4,192,723	0.64	6,565,014	55.57
2019	4,438,619	6,350,419	43.07	4,717,163	6.27	4,345,099	2.10
Average relative error			29.35%		24.99%		17.42%

* All actual and forecast values are in units of one.

**Table 5 sensors-23-03307-t005:** Potential market scale and prospective forecast of field planting sensors.

Sensor Type	Configuration Standard	Potential Market Size (CNY 100 Million)	Market Prospective Forecast	
			In 2025	In 2035
Environmental information sensing sensor	5/6666.7 m^2^	4500	225 million	450 million
Agricultural machinery sensor	30/unit	664	35.23 million	74.7 million

**Table 6 sensors-23-03307-t006:** Potential market scale and prospective forecast of sensors in the facility planting industry.

Sensor Type	Configuration Standard	Potential Market Size (CNY 100 Million)	Market Prospective Forecast	
			In 2025	In 2035
Environmental information sensing sensor	8/666.67 m^2^	540	108 million	168 million
Plant life information sensor	4/666.67 m^2^	270	54 million	84 million

**Table 7 sensors-23-03307-t007:** Potential market scale and prospective forecast of sensors in the livestock breeding industry (pigs).

Sensor Type	Configuration Standard	Potential Market Size (CNY 100 Million)	Market Prospective Forecast	
			In 2025	In 2035
Environmental information sensing sensor	5/20 pigs	412	43.37 million	70.12 million
Animal life information sensor	4/pig	860	86 million	129 million

**Table 8 sensors-23-03307-t008:** Potential scale and prospective forecast of the aquaculture (freshwater) sensor market.

Sensor Type	Configuration Standard	Potential Market Size (CNY 100 Million)	Market Prospective Forecast	
			In 2025	In 2035
Aquaculture environment sensor	4/6666.7 m^2^	152	9.2 million	23 million

## Data Availability

Not applicable.

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
