# Peer review of "Analysis of the Development Status and Prospect of China’s Agricultural Sensor Market under Smart Agriculture"

_sensors, 2023, doi:10.3390/s23063307_

Round 1
Reviewer 1 Report
1-The abstract is not reflected result, material, and another topic. So, it should be rewritten.
2-The manuscript does not like article. It looks like a review. Especially material and method section also rewritten.
3-Discussion and Conclusion is written well. The discussion section was discussed well according to literature.
4-In addition, the literature numbers should be increased because it looks like a review.
Author Response
Thank you very much for your comments and suggestion. The comments are all valuable and very helpful for revising and improving our manuscript. Changes have been made as suggested in the revised version.
- Please we have rewritten the summary section, which can reflects the method, results and main contributions of this article.
Abstract: Agricultural sensors are essensial technologies for smart agriculture, which can transform non-electrical physical quantities such as environmental factors. The ecological elements inside and outside plants and animals are converted into electrical signals for control system recognition, providing a basis for decision-making in smart agriculture. With the rapid development of smart agriculture in China, agricultural sensors have ushered in opportunities and challenges. This paper analyzes the market prospects and scale of agricultural sensors in China based on a literature review and data statistics. The analysis is based on four perspectives of field farming, facility farming, livestock and poultry farming and aquaculture. The demand for agricultural sensors in 2025 and 2035 is also predicted. Based on the findings, the current research affirms that China's sensor market has a good development prospect. However, the critical challenges of China's agricultural sensor industry include a weak technical foundation, poor enterprise research capacity, sensor dependence on imports, and lack of financial support. It is proposed that the agricultural sensor market should be comprehensively distributed in terms of policy, capital, talent, and new technology. In addition, this paper highlighted the future development direction of China's agricultural sensor technology in combination with new technologies and China's agricultural development needs.
- Please according to your sincere suggestion, we have revised the section materials and method.
According to the classification method widely used in China, this study divided China's agricultural sensor market into four categories: open-field farming sensors (environmental information sensing sensors and farm machinery sensors), facility farming sensors (environmental information sensing sensors and plant life information sensors), livestock and poultry farming sensors (pig life sensors only) and aquaculture sensors (freshwater farming).
A literature search was carried out by searching the “China National Knowledge Infrastructure (CNKI)”, “Web of Science”, “Springer Link” and “Science Direct” with the keywords of “smart agriculture”, “agricultural sensors”, “agricultural sensor market”, “agricultural sensor industry”, “development prospects ”,“existing problems” and “China”. The selection of articles was made on based on the keywords mentioned above and included the following criteria: (1) articles related to agricultural sensors and sensor technology; (2) data and forecast on the application of agricultural sensors in China; (3) analysis on the prospect of China's agricultural sensor market and existing problems. The search results were limited to publications in English and Chinese, and priority was given to articles published during 2015–2022. In addition, this paper also searched the government documents and official websites about agricultural sensors in China and consulted the documents related to SA and agricultural sensors (Table 1).
- Please we appreciate your appreciation of the sections discussion and inclusion. We also thank you very much for your valuable suggestions on the improvement of this article.
- Please we have added relevant articles in the sections introduction, state-of-the-art and related work and discussion of the article, which makes the article more scientific. Thank you very much for your suggestions(in lines 49-56, 74-99, 111-113, 116-119 and 158-166).
Reviewer 2 Report
Accept in present form
Author Response
Thank you so much for your comments.
Reviewer 3 Report
Title: “Analysis on the Development Status and Prospect of China's Agricultural Sensor Market under the Smart Agriculture” I have read this Paper thoroughly and have some observations:
1. Conclusion presentation is very poor. Please add future research direction in conclusion section with elaboration.
2. Please highlight main contribution of the research in Abstract section.
3. Please add discussion section.
4. Comparative analysis presentation in the revised paper.
5. Paper has some grammatical errors please re-check all grammatical mistake in the revised version.
6. What is the physical significant of Fig. 2.
7. Introduction section has much non related citation. Please add some related citation.
Multi-criteria decision making of water resource management problem (in Agriculture field, Purulia district) based on possibility measures under generalized single valued non-linear bipolar neutrosophic environment, Expert Systems with Applications.
Author Response
Thank you very much for your detailed comments and suggestions. We found the comments and suggestions helpful. We have revised the paper point-by-point as follows:
- Please we have rewritten the conclusion based on your valuable suggestions, and have detailed the future research and development directions in the conclusion and discussion sections.
Conclusion: “Since agriculture is the backbone of any country, it is necessary to ensure its sustainable growth over the years. The development of SA and sensor technology provides opportunities for the sustainable development of agriculture. Based on literature review, data statistics, and the development trend of China's agriculture, this work predicts the market potential of China's agricultural sensors and points out that China's sensor technology and enterprise R&D strength cannot meet the needs of China's sensor industry. The sensors produced in China have the limitations of single measurement type, poor accuracy, high failure rate, and lack of sensor communication technology support. Similarly, sensor enterprises lack technology accumulation in materials and technology and cannot break through technical barriers.
The current study posits solutions to addressing the abovementioned problems. First, China's government must prioritize agricultural sensors as a principal technology for rural revitalization and development and design top-notch industrial systems. Second, enterprises must establish a sensor industry chain conducive to technology accumulation, research, and innovation. Third, regarding technological breakthroughs, essential components such as sensor chips, wireless sensor networks, micro-integrated sensors in complex environments, and new sensor technologies such as digital compensation technology are the future development direction of sensor technology in China. In addition, this paper also proposes that agricultural sensors will gradually develop towards low cost, high stability, high intelligence, portability, and ease of operation. Future technical improvements are expected to address the limitations of agricultural sensors”.
- Please we have rewritten the Abstract section.
Abstract: Agricultural sensors are essensial technologies for smart agriculture, which can transform non-electrical physical quantities such as environmental factors. The ecological elements inside and outside plants and animals are converted into electrical signals for control system recognition, providing a basis for decision-making in smart agriculture. With the rapid development of smart agriculture in China, agricultural sensors have ushered in opportunities and challenges. This paper analyzes the market prospects and scale of agricultural sensors in China based on a literature review and data statistics. The analysis is based on four perspectives of field farming, facility farming, livestock and poultry farming and aquaculture. The demand for agricultural sensors in 2025 and 2035 is also predicted. Based on the findings, the current research affirms that China's sensor market has a good development prospect. However, the critical challenges of China's agricultural sensor industry include a weak technical foundation, poor enterprise research capacity, sensor dependence on imports, and lack of financial support. It is proposed that the agricultural sensor market should be comprehensively distributed in terms of policy, capital, talent, and new technology. In addition, this paper highlighted the future development direction of China's agricultural sensor technology in combination with new technologies and China's agricultural development needs.
- Please the fifth section of the article is d According to your valuable suggestions, we have expanded the discussion content to make the article more scientific.
China’s agricultural-sensor industry was established in the 1960s. Supported by the rapid development of remote sensing technology, the industry gradually developed. At the end of the 1980s, remote-sensing data acquisition using multiband sensors on satellites was established in China to monitor and evaluate agricultural production. At the same time, the connected-greenhouse structure was introduced, and the scale of enclosed crop cultivation began to expand. Agricultural researchers and practitioners of enclosed crop cultivation began to use sensors in greenhouses and explore the applications of sensors in the agricultural industry. Over the past 40 years, China's agricultural sensor industry has transformed into a developed industrial system. Environmental sensors such as air-temperature, air-humidity and soil-moisture sensors have been commonly used in agricultural production. Advanced technologies in environmental sensors focus on temperature, light, soil moisture, pH, and other indicator measurements. However, there is a lack of sensors for dynamic monitoring of the ecologically integrated environment of farmland and plant growth information. Research on dynamic sensing and monitoring technologies for harmful pollutants such as heavy metals and pesticide residues in soil, essential environmental factors, and kinetic models for plant-soil-environment interactions is still insufficient. There is a lack of methods for highly sensitive, selective, multi-point simultaneous, or multi-component high-throughput detection of the above single-component detection objects . On the other hand, temperature sensors, ammonia sensors, dissolved-oxygen sensors, and pH sensors have been used in aquaculture production. Moreover, multifunctional sensors integrating different types of sensors have been developed. Entering China's "14th Five-Year Plan" period, the agricultural-sensor industry is provided with a significant opportunity for development due to the great demand for agricultural sensors. Despite the vital opportunity, China lags behind technologically advanced countries in research on agricultural sensors and market share in the global market for agricultural sensors. Therefore, comprehensive improvements are required.
In China, research on agricultural sensors is still in its early stage. China produces less than 10 % of the world's agricultural sensors due to a dearth of investment and qualified research and development personnel. Most of the agricultural sensors produced in China are modified industrial sensors. Therefore, agricultural sensors lack accuracy, stability, and stress resistance. Moreover, neither animal nor plant sensors are produced in China. Core algorithms for big data mining process data collected by high-tech agricultural sensors. Autonomous, controllable, intelligent high-tech agricultural sensors have not been produced in China. Most high-tech agricultural sensors developed in China are still in the laboratory testing stage. Significant differences exist in stability, reliability, and power consumption between domestically produced and imported agricultural sensors. Besides, the industrial development mechanism of China's agricultural sensor technology needs to be improved. Problems with the industrial structure of agricultural sensors include fragmentation of enterprises, weakness, low technology level, the existence of a large number of similar products, and lack of innovation. Few companies in China can independently develop and improve sensors, while the majority of sensor companies duplicate or assemble similar products from overseas. As a result, the core technology accumulation of these enterprises is insufficient and far behind the developed countries. For example, the cost of developing mainstream sensor chips based on MEMS technology is high. Even a large-scale professional, scientific research team needs at least ten years of technical accumulation to succeed in research. Small sensor enterprises find it challenging to devote time and financial resources to developing new technology.
However, the United States, Germany, Japan and other countries are leading in agricultural sensors. These countries have a monopoly on sensing components, high-end agricultural environmental sensors, plant and animal life information sensors, and online inspection equipment for the quality of agricultural products and other related technology products. The United States is one of the pioneers in using agricultural sensors in the agriculture sector. Not only does agricultural sensor technology in the United States feature a sophisticated measurement system, but the obtained data is also diverse and accurate. Different types of agricultural sensors have the characteristics of high integration. Therefore, the stability of information transmission between agricultural sensors is relatively strong. In addition, the United States invests more money in researching new technologies and new products of agricultural sensors every year. For example, the United States is developing laser-induced spectroscopy technology for measuring soil nutrients and heavy metal content and using micro and nanotechnology to develop sensors that can enter the metabolic cycle system of plant and animal life forms. China's proportion of the global market for agricultural sensors is considerably less than that of the United States, Japan, and Germany. Agricultural sensors produced in China are manufactured on a small scale with low-level technology. Therefore, the profitability of the sensors is low because of the lack of economies of scale. As a result, approximately 90% of imported high-tech agricultural sensors dominate the Chinese market for agricultural sensors.
To summarize, smart agriculture has become an important way for developed countries to increase the export value of agricultural products. As the technical support of smart agriculture, the industrialization level of agricultural sensors in China lags far behind than that of developed countries, mainly including the following aspects. (a) Large differences in sensor accuracy, low-cost performance, high product failure rate, and low informatization level. (b) Lack of sensor communication technology, such as wireless sensor network communication, heavily affects the accuracy, reliability, and coverage. (c) The measurement type is singular, and the multi-parameter and multi-field integrated sensor needs to be developed urgently. (d) Lack of sensor intelligent calibration technology and application in error analysis, modeling, and processing. (e) Lack of comprehensive and reliable monitoring technology, including electromagnetic compatibility technology and comprehensive monitoring of environmental reliability and safety regulations. (f) Diversity is absent in the types of agricultural planting and breeding sensors, and the application of biological agriculture planting and breeding sensors based on nano and microsystem technology needs to be developed urgently. (g) Sensor materials and processes, such as nanoscale materials and MEMS sensor technologies, lack technological accumulation. China will need to make significant advancements in agricultural sensor technology. First, future optical, electrochemical, electromagnetic, ultrasonic, image, and other methods based on new mechanisms of agricultural sensing, and sensitive devices, photoelectric conversion, weak signal processing, and other core components will become the critical research direction of China's agricultural sensors. Then, new agricultural sensors with micro and small sizes, high reliability, low power consumption, low cost, and long life under complex agricultural environments are also the cardinal direction of sensor technology development in China. According to the analysis of the products exhibited in the International Industrial Exhibition and the technology development of the internationally renowned manufacturers, the development trend of sensor technology is diverse. These encompass digital compensation, networking, intelligent, and multi-function composite technologies. China should adapt its policies, finances, human resources, and other resources and technology to the agricultural sensor market to accomplish its goals. By doing so, this is China's primary strategy to break other countries' monopoly on technical products by developing a batch of agricultural sensors with high precision, low cost, and increased stability.
- Please in the sections introduction and discussion, we revised the comparative analysis of agricultural sensor technology and sensor industry between China and developed countries(in lines 74-99 and 442-483). Thank you very much for your valuable comments.
- Please we have checked the full text of the article for grammatical and other textual problems.
- Please figure 2 is used to show that China has made relevant adjustments to facility agriculture in 2015.The area of facility agriculture has changed significantly in this year. However, after 2015, the development of facility agriculture tends to be stable. Based on this, we have analyzed and predicted the demand and market potential of agricultural sensors in China's facility agriculture in 2025 and 2035.
- Please we have added relevant articles in the sections introduction and state-of-the-art and related work of this article.Thank you very much for your valuable suggestions(in lines 49-56, 74-99, 111-113, 116-119 and 158-166).
Round 2
Reviewer 1 Report
Plz read the manuscript again for correction a few mistakes related words.
Author Response
Thank you for your valuable comments and suggestions on our manuscript. We have carefully revised the full text.
Reviewer 3 Report
The article did not improve with suggestions.
Author Response
Thank you for your valuable comments and suggestions on our manuscript entitled “Analysis on the Development Status and Prospect of China's Agricultural Sensor Market under the Smart Agriculture” (sensors-2241588). The comments are valuable and useful for improving our manuscript and serve an important guiding significance to our research. We have studied them carefully and have made revision in “Track Changes” in the manuscript. Please find our response as follows:
- Please we have rewritten the conclusion based on your valuable suggestions as follows:
“Since agriculture is the backbone of any country, ensuring its sustainable growth over the years is necessary. The development of SA and sensor technology provides opportunities for the sustainable development of agriculture. Based on literature review,data statistics, and the development trend of China's agriculture, this work predicts the market potential of China's agricultural sensors, highlighting that China's sensor technology and enterprise R&D strength cannot meet the needs of China's sensor industry. The sensors produced in China have the limitations of single measurement type, poor accuracy, high failure rate, and lack of sensor communication technology support. Similarly, sensor enterprises lack technology accumulation in materials and technology and cannot break through technical barriers.
The current study posits solutions to addressing the abovementioned problems. First, China's government must prioritize agricultural sensors as a principal technology for rural revitalization and development and design top-notch industrial systems. Second, enterprises must establish a sensor industry chain conducive to technology accumulation, research, and innovation. Third, regarding technological breakthroughs, future optical, electrochemical, electromagnetic, ultrasonic, image, and other methods based on new mechanisms of agricultural sensing, MEMS sensor, digital compensation, networking, intelligent, and multi-function composite technologies, and sensitive devices, photoelectric conversion, weak signal processing, and other core components, will become the critical research direction of China's agricultural sensors. In addition, this paper also proposes that agricultural sensors will gradually develop towards low cost, high stability, high intelligence, portability, and ease of operation. Future technical improvements are expected to address the limitations of agricultural sensors.”
Please we have included the future research direction based on three aspects as methods, technology and core components of agricultural sensors.
Future research would focus on methodology, technology and core components of agricultural sensors (Page14, line 528-533). First, from the development method of agricultural sensor technology, future optical, electrochemical, electromagnetic, ultrasonic, image, and other techniques based on new mechanisms of agricultural sensing will be the research hotspots to improve the accuracy and stability of agricultural sensors. Second, from the perspective of agricultural sensor technology development, the technologies of MEMS sensor, digital compensation, networking, intelligent, and multi-function composite technologies will provide strong support for the further development and application of agricultural sensors in the future. Third, from the perspective of core component composition, such as sensor chips, sensitive devices, photoelectric conversion, weak signal processing, and other core components will become the research direction for upgrading and improving agricultural sensor hardware.
- Please we have highlighted the main contribution of the research in the Abstract section as follows:
“Agricultural sensors are a key technology for smart agriculture, which can transform non-electrical physical quantities such as environmental factors inside and outside plants and animals into electrical signals for control system recognition, providing a basis for decision-making in smart agriculture. With the rapid development of smart agriculture in China, agricultural sensors have ushered in opportunities and challenges. Based on literature review and data statistics, this paper analyzes the market prospects and market scale of agricultural sensors in China from four perspectives: field farming, facility farming, livestock and poultry farming and aquaculture. The study further predicts the demand for agricultural sensors in 2025 and 2035. The results reveal that China's sensor market has a good development prospect. However, the paper garnered that the key challenges of China's agricultural sensor industry include a weak technical foundation, poor enterprise research capacity, high importation of sensors and lack of financial support. Given this, the agricultural sensor market should be comprehensively distributed in terms of policy, funding, expertise, and innovative technology. In addition, this paper highlighted integrating the future development direction of China's agricultural sensor technology with new technologies and China's agricultural development needs.”
- Please we have added thediscussion section in chapter 5. In this section, we discussed the development history of agricultural sensors in China, the development of the sensor industry, and the current problems. In addition, we also pointed out the gap between China's agricultural sensors and developed countries in sensor technology and industrial development in this section. The discussion section is as follows (Page 11-13: lines 411-512) :
“China’s agricultural-sensor industry was established in the 1960s [53]. Supported by the rapid development of remote sensing technology, the industry gradually developed. At the end of the 1980s, remote-sensing data acquisition using multiband sensors on satellites was established in China to monitor and evaluate agricultural production. At the same time, the connected-greenhouse structure was introduced, and the scale of enclosed crop cultivation began to expand. Agricultural researchers and practitioners of enclosed crop cultivation began to use sensors in greenhouses and explore the applications of sensors in the agricultural industry. Over the past 40 years, China's agricultural sensor industry has transformed into a developed industrial system [53]. Environmental sensors such as air-temperature, air-humidity and soil-moisture sensors have been commonly used in agricultural production. Advanced technologies in environmental sensors focus on temperature, light, soil moisture, pH, and other indicator measurements. However, there is a lack of sensors for dynamic monitoring of the ecologically integrated environment of farmland and plant growth information. Research on dynamic sensing and monitoring technologies for harmful pollutants such as heavy metals and pesticide residues in soil, essential environmental factors, and kinetic models for plant-soil-environment interactions is still insufficient. There is a lack of methods for highly-sensitive, selective, multi-point simultaneous, or multi-component high-throughput detection of the above single-component detection objects [54-55]. On the other hand, temperature sensors, ammonia sensors, dissolved-oxygen sensors, and pH sensors have been used in aquaculture production. Moreover, multifunctional sensors integrating different types of sensors have been developed. Entering China's "14th Five-Year Plan" period, the agricultural-sensor industry is provided with a significant opportunity for development due to the great demand for agricultural sensors. Despite the vital opportunity, China lags behind technologically advanced countries in research on agricultural sensors and market share in the global market for agricultural sensors. Therefore, comprehensive improvements are required.
In China, research on agricultural sensors is still in its early stage. China produces less than 10 % of the world's agricultural sensors due to a dearth of investment and qualified research and development personnel [56]. Most of the agricultural sensors produced in China are modified industrial sensors. Therefore, agricultural sensors lack accuracy, stability, and stress resistance. Moreover, neither animal nor plant sensors are produced in China. Core algorithms for big data mining process data collected by high-tech agricultural sensors [57]. Autonomous, controllable, intelligent high-tech agricultural sensors have not been produced in China. Most high-tech agricultural sensors developed in China are still in the laboratory testing stage. Significant differences exist in stability, reliability, and power consumption between domestically produced and imported agricultural sensors [58]. Besides, the industrial development mechanism of China's agricultural sensor technology needs to be improved. Problems with the industrial structure of agricultural sensors include fragmentation of enterprises, weakness, low technology level, the existence of a large number of similar products, and lack of innovation. Few companies in China can independently develop and improve sensors, while the majority of sensor companies duplicate or assemble similar products from overseas. As a result, the core technology accumulation of these enterprises is insufficient and far behind the developed countries. For example, the cost of developing mainstream sensor chips based on MEMS technology is high. Even a large-scale professional, scientific research team needs at least ten years of technical accumulation to succeed in research. Small sensor enterprises find it challenging to devote time and financial resources to developing new technology.
However, the United States, Germany, Japan and other countries are leading in agricultural sensors. These countries have a monopoly on sensing components, high-end agricultural environmental sensors, plant and animal life information sensors, and online inspection equipment for the quality of agricultural products and other related technology products. The United States is one of the pioneers in using agricultural sensors in the agriculture sector. Not only does agricultural sensor technology in the United States feature a sophisticated measurement system, but the obtained data is also diverse and accurate. Different types of agricultural sensors have the characteristics of high integration. Therefore, the stability of information transmission between agricultural sensors is relatively strong. In addition, the United States invests more money in researching new technologies and new products of agricultural sensors every year. For example, the United States is developing laser-induced spectroscopy technology for measuring soil nutrients and heavy metal content and using micro and nanotechnology to develop sensors that can enter the metabolic cycle system of plant and animal life forms [59]. China's proportion of the global market for agricultural sensors is considerably less than that of the United States, Japan, and Germany. Agricultural sensors produced in China are manufactured on a small scale with low-level technology. Therefore, the profitability of the sensors is low because of the lack of economies of scale. As a result, approximately 90% of imported high-tech agricultural sensors dominate the Chinese market for agricultural sensors [10].
To summarize, smart agriculture has become an essential way for developed countries to increase the export value of agricultural products. As the technical support of smart agriculture, the industrialization level of agricultural sensors in China lags far behind than that of developed countries, mainly including the following aspects. (a) Large differences in sensor accuracy, low-cost performance, high product failure rate, and low informatization level. (b) Lack of sensor communication technology, such as wireless sensor network communication, heavily affects the accuracy, reliability, and coverage. (c) The measurement type is singular, and the multi-parameter and multi-field integrated sensor needs to be developed urgently. (d) Lack of sensor intelligent calibration technology and application in error analysis, modeling, and processing. (e) Lack of comprehensive and reliable monitoring technology, including electromagnetic compatibility technology and comprehensive monitoring of environmental reliability and safety regulations. (f) Diversity is absent in the types of agricultural planting and breeding sensors. Also, applying biological agriculture planting and breeding sensors based on nano and microsystem technology must be urgently developed. (g) Sensor materials and processes, such as nanoscale materials and MEMS sensor technologies, lack technological accumulation. China will need to make significant advancements in agricultural sensor technology. First, future optical, electrochemical, electromagnetic, ultrasonic, image, and other methods based on new mechanisms of agricultural sensing, and sensitive devices, photoelectric conversion, weak signal processing, and other core components will become the critical research direction of China's agricultural sensors. Then, new agricultural sensors with micro and small sizes, high reliability, low power consumption, low cost, and long life under complex agricultural environments are also the cardinal direction of sensor technology development in China. According to the analysis of the products exhibited in the International Industrial Exhibition and the technology development of the internationally renowned manufacturers, the development trend of sensor technology is diverse. These encompass digital compensation, networking, intelligent, and multi-function composite technologies. China should adapt its policies, finances, human resources, and other resources and technology to the agricultural sensor market to accomplish its goals. By doing so, this is China's primary strategy to break other countries' monopoly on technical products by developing a batch of agricultural sensors with high precision, low cost, and increased stability.”
- Please we have added a comparative analysis presentation from the perspective of agricultural sensor technology and industrial development between China and developed countries in the discussion section
Page12: lines 438-440: “China produces less than 10 % of the world's agricultural sensors due to a dearth of investment and qualified research and development personnel ” , Page13: lines 474-477: “China's proportion of the global market for agricultural sensors is considerably less than that of the United States, Japan, and Germany”.
Page12: lines 446-448: “Significant differences exist in stability, reliability, and power consumption between domestically produced and imported agricultural sensors ”, Page12: lines 440-442: “Most of the agricultural sensors produced in China are modified industrial sensors. Therefore, agricultural sensors lack accuracy, stability, and stress resistance”, Page12: lines 465-469: “Not only does agricultural sensor technology in the United States feature a sophisticated measurement system, but the obtained data is also diverse and accurate. Different types of agricultural sensors have the characteristics of high integration. Therefore, the stability of information transmission between agricultural sensors is relatively strong”.
Page12: lines 449-454: “Problems with the industrial structure of agricultural sensors include fragmentation of enterprises, weakness, low technology level, the existence of a large number of similar products, and lack of innovation. Few companies in China can independently develop and improve sensors, while the majority of sensor companies duplicate or assemble similar products from overseas ”, Page12: lines 460-464: “However, the United States, Germany, Japan and other countries are leading in agricultural sensors. These countries have a monopoly on sensing components, high-end agricultural environmental sensors, plant and animal life information sensors, and online inspection equipment for the quality of agricultural products and other related technology products” , Page12: lines 464-467: “Not only does agricultural sensor technology in the United States feature a sophisticated measurement system, but the obtained data is also diverse and accurate”.
- Please we have re-checked and revised grammatical errors and other textual or grammatical errors in all chapters of this paper. Thank you very much for your valuable suggestion.
Examples:
Page1: line 43-45: “The sensors' accumulated data about the object or environment can be used to identify people, location, objects and their states, which is a guiding principle for scientific research”.
Page2, lines 142-145: “Currently, more wearable sensor technologies such as motion sensors and vital signs sensors are used within the breeding environment to collect, transmit, and store information”.
Page5, line 210-211: “In China’ s agricultural industry, tillage, seeding, pipe installation, and harvesting machines have sensors to monitor their operation”.
Page6, line 238: “Parameter estimate obtained by logarithmic transformation”.
Page9, line 316-319: “the collecting, storing, and analyzing of data related to farming processes. The 526 animal-husbandry enterprises surveyed in 2020 revealed that 54.55 % had automated monitoring of breeding environments”.
- Please the significance of Figure 2 is that the planting area of China's facility agriculture has entered a stable development state after the adjustment in 2015. In this regard, representing this in Figure 2 can provide a clearer overview of the development over the years (2013-2019). The figure serves as the basis for analyzing and predicting the demand from the enclosed-crop-cultivation industry for agricultural sensorsin 2025 and 2035.
Data source: China County Statistical Yearbook, National Bureau of Statistics
Figure 2. Changes in planting area of facility agriculture in China from 2013 to 2019
- Thank you very much for your valuable suggestions.Please we have deleted non-related citations and added relevant articles as follows:
Deleted non-related citations:
These sentences were deleted in page2, lines 48-61 and page1, line:44-46.
“ A rural-revitalization strategy was proposed at the 19th National Congress of the Communist Party of China (CPC) to modernize the country’s economy. Since then, the Central Committee of the CPC and the State Council have emphasized the great importance of SA. The No. 1 Central Document for 2021 restated the necessity of developing SA that integrates information technology and agricultural production in establishing agricultural and rural big-data systems. According to the Report on Digital Development in Rural China 2020, the construction of information infrastructure in rural areas in China has shown continuous improvement. Fiber optics and 4G coverage have been available in 98 % of administrative villages. The innovative applications of next-generation technologies (artificial intelligence, 5G technology, and big data) have been steadily promoted. Supported by information technology and plant-breeding technology, SA, along with supporting policies and solid technical foundations, will promote the development of China's agricultural industry and help realize modern agriculture in China in 2035”.
“By integrating modern information technologies, SA realizes intelligent control, precise input, and personalized service. As an agricultural breakthrough after plant breeding, SA can promote the modernization of the agricultural industry and improve the industry’s economic development and environmental sustainability.”
Added relevant articles in the Introduction section and revised the references accordingly (Page2: lines 70-95):
“ The United States, Japan, and Germany hold the largest market share in the global sensor market distribution. Whiles other countries, including China, lag behind with a significant gap in market share [7]. For instance, developed countries, including Europe, the United States, and Japan, monopolize the airborne-to-earth observation core sensor technology. In addition, countries such as the United States, Japan, the United Kingdom, France, and Germany have made sensor technology one of their national priorities for developing critical technologies. This has contributed to the rapid development of sensor technology [8]. At present, China's agricultural sensor field is facing many problems such as lack of enterprise technology innovation capability, key technologies, international competitiveness, and unsound industrial ecology [9]. Almost 100% of the medium and high-quality sensing components and 90% of the sensor chips used in China are imported [10]. Due to the lack of high-quality optical components, high-end sensor chips, and other principal technologies and processes, the development of airborne spectral imaging sensors is still in its infancy in China [11]. The research and development of multispectral camera sensors in China are only in the scientific research stage. In this regard, users of agricultural UAV sensors can only use expensive imported multispectral sensors. Similarly, wearable plant water sensors [12], electronic noses for monitoring volatile organic compounds released from plant leaves [13][14], miniature biosensors for detecting pH changes in plant foliar tissues [15], nanosensors for detecting crop stress signaling molecules [16], and other plant and animal life ontology information sensors have a long way to go before they can be widely adopted in China. Chinese agricultural sensors, particularly intelligent agricultural sensors, are heavily dependent on imports. Meanwhile, the production and application of self-produced agricultural sensor core components are insufficient to meet the rapid development of the agricultural IoT and SA [17] ”.
- Yang, Z. The agricultural industrial management mode and development counter measures. Master’s Thesis,Chinese Academy of Agricultural Sciences, Beijing, China, 2015.
- Guo, P.; Huang J.; Xie Q. Application of sensors in precision agriculture. Guangdong Agricultural Sciences 2014, 41, 5. https://doi.org/10.16768/j.issn.1004-874x.2014.05.050
- Intelligent sensor spectrum system and development strategy white paper. Available online: http://www.cesi.cn/201908/5426.html(accessed on 25 February 2023).
- Zhang, B.; Zha, Y.; Shi, The status, potential risks and countermeasure proposals of import-dependent on intelligent agricultural equipment. China Agricultural Informatics 2019, 31, 113-120. DOI: CNKISUN:NXTS.0.2019-04-014
- Wang, J.; Wang, J.; Zhang, Z. Research Progress on Unmanned Aerial Vehicle for Ecological Remote Sensing Monitoring Based on Bibliometric Assessment. Tropical Geography 2019, 4, 616-624. https://doi.org/10.13284/j.cnki.rddl.003157
- Lan, L. In-situ and non-destructive sensing of plant water information based on self-powered flexible and wearable devices. Doctor’s Thesis,Zhejiang University, Hangzhou, China, 2021.
- Xiao, W.; Tang, X.; Ren Z. Rapid diagnosis ofpesticide residues in fresh tea leaves based on electronic nose technology. Journal of Tea Communication 2021, 3, 484-493. https://doi.org/10.3969/j.issn.1009-525X.2021.03.016
- He, L.; Yi Y.; Peng Y. Analysis of the volatile flavor substances of Chinese prickly ash leaf in different growth stages based on E-nose and gas chromatographymass spectrometry. Journal of Southern Agriculture 2019, 3, 641-648. https://doi.org/3969/j.issn.2095-1191.2019.03.28
- Zou, Q. Microbiosensor and its application on the transgenic plant physiological process in real time. Master’s Thesis,Huazhong University of Science and Technology, Wuhan, China, 2007.
- Li, J.; Wu, H.; Tao Y. Advances in smart agriculture research: application of nanobiotechnology to obtain crop information. Journal of Smart Agriculture 2021, 11, 1-6.
- Application and market of agricultural sensor technology in China: current status and future perspectives.Available online: http://kns.cnki.net/kcms/detail/33.1247.S.20220906.1417.004.html(accessed on 25 February 2023).
